# Targeting DNA Replication Stress and DNA Double-Strand Break Repair for Optimizing SCLC Treatment

**DOI:** 10.3390/cancers11091289

**Published:** 2019-09-02

**Authors:** Xing Bian, Wenchu Lin

**Affiliations:** 1High Magnetic Field Laboratory, Chinese Academy of Sciences, Hefei 230031, Anhui, China; 2University of Science and Technology of China, Hefei 230026, Anhui, China; 3Key Laboratory of High Magnetic Field and Ion Beam Physical Biology, Hefei Institutes of Physical Science, Chinese Academy of Sciences, Hefei 230031, Anhui, China

**Keywords:** small cell lung cancer, replication stress response, DSD repair, therapeutic strategy

## Abstract

Small cell lung cancer (SCLC), accounting for about 15% of all cases of lung cancer worldwide, is the most lethal form of lung cancer. Despite an initially high response rate of SCLC to standard treatment, almost all patients are invariably relapsed within one year. Effective therapeutic strategies are urgently needed to improve clinical outcomes. Replication stress is a hallmark of SCLC due to several intrinsic factors. As a consequence, constitutive activation of the replication stress response (RSR) pathway and DNA damage repair system is involved in counteracting this genotoxic stress. Therefore, therapeutic targeting of such RSR and DNA damage repair pathways will be likely to kill SCLC cells preferentially and may be exploited in improving chemotherapeutic efficiency through interfering with DNA replication to exert their functions. Here, we summarize potentially valuable targets involved in the RSR and DNA damage repair pathways, rationales for targeting them in SCLC treatment and ongoing clinical trials, as well as possible predictive biomarkers for patient selection in the management of SCLC.

## 1. Introduction

Small cell lung cancer (SCLC) is a deadly neuroendocrine tumor that accounts for 15% of all lung cancers [1,2]. SCLC distinguishes clinically from non-small cell lung cancer by its rapid growth and early distant metastases [3]. During the past three decades, the treatment strategy for SCLC is extremely limited, and chemotherapy remains the cornerstone of therapeutic options for SCLC patients [4]. In mammalian cells, the maintenance of genome integrity requires faithful DNA duplication during cell cycle progression [5,6]. To transmit DNA precisely to daughter cells, cells have evolved coordinated and interweaved molecular networks to antagonize the genotoxic stress generated from intrinsic factors and extrinsic causes [7,8,9]. The defects in the restriction and G1 checkpoints promote the G1-S transition in SCLC cells, leading to premature onset of the S phase [10,11]. Amplification of oncogenes such as *MYC* family genes and activation of signal transduction cascades like the PI3K/AKT pathway in SCLC also drive rapid cell proliferation [12]. Promotion of S phase entry and demands of unrestrained proliferation often interfere with DNA replication and induce stalled replication fork termed replication stress (RS; Figure 1) [13]. When the impediments to DNA replication progression cannot be handled in time, the stalled replication forks are susceptible to fork collapse, leading to highly lethal DNA double-strand breaks (DSBs) [14,15]. To prevent the cytotoxicity caused by replication stress, SCLC cells develop a robust DNA damage response (DDR) network including an effective replication stress response (RSR) pathway and a constitutive DNA repair system to tolerate high levels of RS and to reduce consequent DNA damage [16] (Figure 2). However, the high RS level and the strong DNA damage response in SCLC create a vulnerability to genome integrity. Indeed, SCLC often presents with a super high mutation burden [12]. Thus, a coherent understanding of how SCLC cells manipulate replication stress response (RSR) to control DNA replication and to fix damaged DNA may facilitate the development of new therapeutic strategies and circumvent drug resistance in the SCLC treatment. Recent advances in transcriptomics and proteomics have identified a significantly elevated expression of a number of genes, which encode vital proteins responsible for DNA damage response in SCLC including *ATR*, *CHK1*, *WEE1*, and *BRCA1* [17]. In recent years, a great amount of effort has been put on the discovery and development of compounds that would exploit defects in DNA replication and DNA repair to treat cancer [18]. However, although the single therapeutic agents to target DNA damage response have shown promising effects, the drug-resistance is often observed due to the complexity of the DNA damage response network [19,20]. Therefore, to expand the therapeutic efficacy, combinations of replication stress inducers with other therapeutics have been investigated in preclinical and clinical studies and have shown augmented beneficial effects compared with either agent alone [21,22,23,24].

Here, we provide an overview of the major sources of replication stress in SCLC cells and the signaling pathways responsible for DDR. We then outline the pharmacological approaches to exacerbate replication stress and to target DNA damage repair for SCLC treatment, with a specific focus on RSR and DSB repair. Lastly, we discuss the combination strategies for treatment and propose the potential strategies to further augment treatment efficacy for SCLC.

## 2. Sources of Replication Stress in SCLC

DNA replication stress, arising from endogenous sources, is a hallmark of SCLC [16]. Accumulating evidence indicates that several mechanisms contribute to replication stress in SCLC.

Loss of tumor suppressors and activation of oncogenes are emerging sources of replication stress (Figure 1). Several lines of evidence indicate that tumor suppressors p53 and RB1 are inactivated in nearly 100% of SCLC [11,12,25]. Both p53 and RB1 play crucial roles during cell cycle progression. Given that RB1 associates with the E2F family to suppress cell proliferation, mutated RB1 cannot bind to the E2F family, and the release of E2F family transcription factors activates its downstream targets to degrade the restriction point and promote early initiation of DNA replication [26,27]. In addition, inactivation of p53 leads to loss of activity of G1 checkpoint, which favorites G1-S transition. MYC family members, including *MYC*, *MYCN*, and *MYCL*, are exclusively amplified or overexpressed in a subset of SCLC tumors [12]. MYC family members act as transcription factors (TFs) to promote the S phase entry through promoting increased replication initiation and origin firing [10,28]. Premature onset of S phase or constitutive activation of oncogenes often leads to defects in nucleotide biosynthesis and the DNA replication machinery [29]. 

Abnormal cellular metabolism causes replication stress [30,31]. Cancer cells often rewire metabolic flux to generate metabolites as either a direct or indirect consequence of activation of oncogenic pathways such as PI3K/AKT, which meet the demands for sustained proliferation and many other fundamental cellular functions in SCLC cells [32]. As metabolites such as deoxynucleoside triphosphates (dNTPs) are the basic units for DNA replication, any disturbance of nucleotide anabolism may interfere with DNA replication machinery [33]. The accumulation of reactive oxygen species (ROS) in cancer cells, a common phenomenon in cancer cells, also results in the generation of oxidized nucleotides, which would stall the replication fork [34].

As DNA replication occurs in the context of chromatin, any perturbation of chromatin dynamics may evoke replication stress [35]. Nucleosome, the basic unit of chromatin, is formed by wrapping 146 base-pair of DNA around an octamer consisting of pairs of each of the four core histones (H2A, H2B, H3, and H4) [36]. The tails of histone, especially histone H3 and H4, are subject to post-translational modifications, like methylation, acetylation, phosphorylation, ubiquitylation, and sumoylation. Aberrant chromatin structure mediated by histone modifications and ATP-dependent remodeling may affect chromatin accessibility to nuclear proteins such as DNA polymerases, thereby impeding DNA replication progression, RSR, and DNA repair [37,38]. A number of chromatin modifiers and a range of chromatin-associated changes have been observed at and near sites of replication forks, confirming an indispensable role of chromatin structure in the DNA replication machinery [39]. Furthermore, chromatin modifiers also affect the expression of DNA-replicative enzymes by transcriptional regulation.

## 3. DNA RSR and DSB Repair

If the stalled replication forks are sufficiently prolonged, static extended stretches of single-strand DNA (ssDNA) can be generated by the replicative mini-chromosome maintenance (MCM) [40,41]. In response to replication stress, replication protein A (RPA) is the first one to be loaded onto the unstable ssDNA, and the long stretches of RPA-coated ssDNA adjacent to double-strand DNA (dsDNA) act as a platform to trigger the ATR/CHK1 RSR signaling pathway with the assistance of a number of mediator proteins including TOPBP1 and ETAA1 [29,42,43] (Figure 2). Upon activation, CHK1 dissociates from Claspin to phosphorylate a number of regulators of cell division including phosphatase CDC25A and CDC25C, leading to the arrest of cell cycle progression either in the S-phase or at the boundary of G2/M [29,44]. On the other hand, the ATR-CHK1 pathway suppresses excess late origin firing by multipronged actions. Together, these events stabilize the stalled replication forks and prevent the exhaustion of dNTP pools and replication factors, providing cells with the necessary time to resolve the persistent replication stress [44,45,46].

When the stalled replication fork cannot be stabilized, fork then collapse into a double-strand break, which is the most lethal type of DNA damage [47]. In response to DSBs, the MRE11-RAD50-NBS1 (MRN) complex recruits ATM to damaged DNA sites and stimulates ATM kinase activity [48]. ATM then targets multiple substrates such as the downstream effector kinase CHK2 to induce cell cycle arrest, and to initiate DNA repair. Two major DNA repair pathways are responsible for DSB repair: Faithful homologous recombination (HR) and error-prone non-homologous end joining (NHEJ) [18,49,50,51]. Taken together, targeting RSR or DSB repair may provide an avenue for improving SCLC treatment efficacy (Figure 3).

## 4. Current Chemotherapy for Small Cell Lung Cancer Patients and Its Limitation

Chemotherapy regime of cisplatin or carboplatin plus etoposide remains the major first-line treatment option for SCLC patients with both limited and extensive-stage disease [4,52]. Chemotherapeutic agents currently used in SCLC achieve therapeutic effects primarily through directly or indirectly boosting replication stress [53]. Platinum-based compounds (cisplatin and carboplatin) generate adducts on purine residues, thus resulting in the formation of inter-strand or intra-strand crosslinks [54,55]. These crosslinks will physically block DNA replication progression. Mechanistically, TOP2 cleaves double-stranded DNA to allow passage of an intact DNA duplex through the TOP2-linked DSB [56]. Etoposide, a TOP2-specific inhibitor, stabilizes TOP2 cleavage complexes, thus inducing DSBs and subsequent cell death if not repaired [57]. Besides, TOP1 reduces DNA torsional stress created by twisting and supercoiling during DNA replication progression. Topotecan, the drug approved by the FDA for second-line treatment of SCLC, traps TOP1 cleavage complexes to prevent repair of single-strand breaks. Although SCLC patients do benefit from the conventional chemotherapy strategy, the long-term outcomes remain poor. Additionally, traditional chemotherapeutic agents used for SCLC patients in the clinic have achieved their therapeutic efficacy, especially in cancer cells with the highly proliferative feature. However, the characteristics of these DNA damage agents will kill both cancer cells and normal replicating cells such as hematopoietic progenitor cells, therefore resulting in severe side-effects during the treatment of SCLC patients. Together, the development of novel therapeutic strategies by specifically targeting the RSR pathway and the DSB repair system should not only hope to augment therapeutic effects but also result in fewer side-effects during the clinical treatment of SCLC [16].

## 5. The Rationales for Enhancing Replication Catastrophe in SCLC

In recent years, exacerbating replication stress seems to be a powerful means to kill cancer cells through mitotic catastrophe due to intolerable levels of replication stress [46]. In dealing with high levels of endogenous or exogenous replication stress, SCLC cells have acquired an intricate genome-protective mechanism to counteract high levels of replication stress during the long history of cellular evolution [47]. Transcriptomic profiling analysis uncovered that a number of genes involved in replication fork stabilization are overexpressed in SCLC cells, which might mediate replication stress tolerance [58]. Besides, ATR/CHK1-dependent replication checkpoint is more active in SCLC cells than that in NSCLC cells. More importantly, loss of function of p53 in SCLC cells causes a defective G1 checkpoint, which makes SCLC cells more rely on the intra-S and G2/M checkpoints. In one word, the dependency of replication stress-related proteins in SCLC cells provides potential therapeutic opportunities to harness replication stress to treat SCLC. Indeed, conventional therapeutic agents such as DNA crosslinkers and topoisomerase inhibitors for SCLC achieve their treatment efficacy by perturbation of DNA replication progression. In the last decade, emerging ways to enhance replication stress have shown great potential for SCLC treatment. In this section, we summarize recent advances in utilizing replication stress inducers for the treatment of SCLC.

### 5.1. Targeting Poly (ADP-Ribose) Polymerase (PARP) as a Therapeutic Option for SCLC

PARP family members are multifunctional proteins. One of the major functions of PARP1 is to repair DNA single-strand breaks (SSBs) through base excision repair (BER) [59]. Meanwhile, PARP1 also plays an indispensable role in DSB repair pathways, including HR and NHEJ [60]. Inhibition of PARP in BRCA1/2-deficient breast cancer cells leads to “synthetic lethality”. PARP inhibitors, including olaparib, niraparib, recuparib, and talazoparib have been approved by the FDA for treating advanced breast cancer and ovarian cancer based on the concept of “synthetic lethality” [61,62,63]. Interestingly, a previous study revealed that PARP recruits and collaborates with MRE11 at stalled replication forks to mediate restart of replication forks [64]. Additionally, PARP inhibitors evoke strong CHK1 phosphorylation in several types of cancers [65,66]. Together these studies demonstrate a blockade of PARP heightens replication stress and implicate PARP as a potent therapeutic target for SCLC.

PARP1 is an attractive therapeutic target in SCLC at least partially due to its high expression [17]. PARP inhibitors, including olaparib and recuparib, are highly effective in treating SCLC preclinical models. Besides, a high potent PARP trapper talazoparib exhibits strikingly single-agent activity in SCLC [67]. Of note, the sensitivity of PARP inhibitors in SCLC is associated with the expression of a number of DNA damage repair proteins and the activity of the PI3K/AKT pathway [67]. Since DDR proteins are crucial for maintaining SCLC proliferation, targeting either the RSR pathway or DSB repair system might sensitize SCLC cells to PARP inhibitor. Interestingly, SLFN11, a replication stress sensor, has been demonstrated as a biomarker for predicting the sensitivity of PARP inhibitor talazoparib in SCLC, further emphasizing perturbation of RSR pathway sensitizes SCLC to PARP inhibition [68]. Based on the above preclinical studies, currently, PARP inhibitor olaparib as a monotherapy is being evaluated in phase II clinical trials in SCLC (NCT03009682). Meanwhile, the clinical trial of PARP inhibitor talazoparib as a monotherapy has shown encouraging activity in the treatment of SCLC (NCT01286987).

### 5.2. Targeting ATR/CHK1 Signaling Cascade in SCLC

The ATR/CHK1 signaling cascade plays a pivotal role in suppressing replication stress [69]. Excessive replication stress is harmful to cancer cells as it causes the failure of the mitosis progression and undergoes mitotic catastrophe. The activation of ATR/CHK1 signaling cascade avoids the accumulation of excessive ssDNA and preserves genome integrity [70]. CHK1, an essential mediator of DNA damage-induced cell cycle arrest in the S phase and G2 phase, is highly expressed in SCLC, indicating that SCLC might depend on a robust RSR to confront high genotoxic stress [71]. Indeed, CHK1 inhibitor prexasertib as a single-agent has shown remarkable anti-tumor activity in a preclinical study [21]. A transcriptomic profile found that the expression of CDC25 is much higher in SCLC than that in NSCLC [22]. Pharmacological inhibition of SCLC cells by ATR inhibitor VE-822 and CHK1 inhibitor PF-477736 selectively eliminates SCLC cells, but not NSCLC cells [22]. Additionally, ATR/CHK1 signaling cascade also can be activated by DSB. Therefore, the anti-tumor activity of ATR and CHK1 inhibitors might be due to both compromised RSR and a blockade of DSB repair. Further analysis identifies MYC as a predictive biomarker of sensitivity to CHK1 inhibitors in SCLC. Clinical evaluation of therapeutic efficacy of CHK1 inhibitors in SCLC has shown promising anti-tumor effects (NCT02735980).

### 5.3. WEE1 as a Therapeutic Target in the Treatment of SCLC

The WEE1 kinase serves as a gatekeeper through regulating both S-phase and G2/M phase checkpoints [72,73]. Although WEE1 is not the core component of the RSR pathway, activation of WEE1 by CHK1 targets CDK1 and CDK2 to slow down cell cycle progression upon DNA lesions. Thus, the WEE1 inhibitor may force cells to enter mitosis in the presence of incomplete DNA replication, which might trigger replication stress due to aberrations of replication dynamics and dNTP supply [74]. Furthermore, in cells deficient for p53, WEE1 is important for maintaining genome integrity. Therefore, targeting G2/M checkpoint may provide therapeutic opportunities for SCLC patients. Of note, preclinical studies have shown that the majority of SCLC cell lines are sensitive to WEE1 inhibitor AZD1775 [75]. Further investigation based on SCLC circulating tumor cell-derived explant (CDX) models demonstrated that *MYCL*-induced replication stress and defects in HR repair sensitize SCLC cells to the WEE1 inhibition [76]. Additionally, another study showed that inhibition of AXL or mTOR confers deficiency in DNA repair, and thus enhances the efficacy of WEE1 inhibitor in SCLC [75]. Based on these results, the efficiency of the WEE1 inhibitor and the candidate biomarkers for patient selection in relapsed SCLC is under clinical investigation (NCT02482311; NCT02593019). Besides, phase II study of WEE1 inhibitor AZD1775 as a monotherapy in relapsed SCLC with *MYC* family amplification is underway to evaluate its therapeutic efficacy (NCT02688907).

## 6. The Rationales for Targeting Homologous Recombination Pathway and Non-Homologous End Joining Pathway in the Treatment of SCLC

Although alterations in DSB repair genes in SCLC have been found rarely by a comprehensive genomic profile, proteomic and transcriptomic studies have identified that DNA repair proteins tend to be overexpressed in SCLC [17,58]. In cancer cells with persistent replication stress, failure to stabilize the stalled forks may lead to DSB formation. SCLC cells may drive reliance on a constitutive DSB repair system, which mostly depends on HR and NHEJ repair pathways to repair lethal DNA lesions for cell survival (Figure 2). Therefore, the crucial role of the DSB repair system in SCLC cells constitutes a vulnerability to be exploited therapeutically.

### 6.1. Targeting the Core Components of the DSB Repair Pathway for SCLC Treatment

ATM functions as an apex transducer in response to DSBs. In recent years, ATM emerges as a therapeutic target for cancer treatment [77,78]. A number of ATM inhibitors have been developed, and the preclinical studies of these inhibitors have shown potential anti-tumor effects in several types of cancers [79]. RAD51, a core component of the HR pathway, is another therapeutic candidate that can be targeted for cancer treatment. It has been shown that RAD51 is required for etoposide-induced DSB repair in SCLC, and high expression of RAD51 is associated with radioresistance in a SCLC preclinical model [80]. DNA-dependent protein kinase (DNA-PK), a serine/threonine protein kinase, plays a central role in the NHEJ pathway. Similarly, DNA-PK has been found to contribute to chemoresistance in SCLC [81]. Given that high expression of key components of DSB repair pathways in SCLC, it is conceivable to test the therapeutic effects of targeting the crucial proteins involved in DSB repair pathways in SCLC.

### 6.2. PLK1 as a Therapeutic Target in the Treatment of SCLC

Besides the core components of the DSB repair system, several cell cycle regulators could also be targeted since they play indispensable roles in DNA damage checkpoints [82]. PLK1, one of five members of the family of PLKs, is traditionally recognized as a key cell cycle regulator [83]. Accumulating evidence indicates that PLK1 is involved in DNA damage response and G2 DNA damage recovery pathway through regulation of CDC25C and WEE1 [84]. Although the therapeutic benefit of a single-agent PLK1 inhibitor has been obtained in SCLC preclinical models, a clinical trial has shown that relapsed SCLC was not responsive to single-agent PLK1 inhibitor BI2536 [85,86]. Therefore, new therapeutics that expands the utility of the PLK1 inhibitor and identification of biomarkers for patient selection in SCLC warrants further studies. 

### 6.3. Aurora Kinase as a Therapeutic Target in the Treatment of SCLC

Aurora kinases are regarded as key cell cycle mediators through regulating G2/M DNA damage checkpoint and other mitotic events [87]. Mechanistically, Aurora kinase A which is encoded by *AURKA* promotes mitosis entry through phosphorylating and activating PLK1 [88]. Aurora kinase B serves as a component of the chromosomal passenger complex to regulate chromosome segregation and cytokinesis in mitosis [89]. The gene encoding Aurora kinase A is frequently amplified in various cancer types, including SCLC, highlighting it as a potent therapeutic target in cancer treatment. A recent study showed that AURKA/TPX2 acts as a heterodimer to protect stalled replication forks after DNA damage [90]. Loss of *AURKA* compromises DNA end resection, thus, decreasing homologous recombination activity [90]. In ovarian cancer, inhibition of Aurora kinase A stimulates NHEJ signaling activity by increasing phosphorylated DNA-PK and reduces HR activity through decreasing PARP and BRCA1/2 expression [91]. Indeed, several preclinical studies have demonstrated the therapeutic efficacy of Aurora kinase inhibitors in SCLC [92,93,94]. Further investigation indicates that SCLC cells with *MYC* amplification or *MYC* overexpression are significantly more responsive to Aurora kinase inhibitors such as alisertib and barasetib than *MYC* family gene-non-amplified SCLC. A clinical phase II study of alisertib in combination with paclitaxel obtained an impressive response rate compared with placebo in combination with paclitaxel in SCLC patients (NCT02038647). Besides, additional Aurora kinase inhibitors are under clinical development to assess its therapeutic activity (NCT03216343; NCT03898791; NCT03092934).

## 7. Using Drug Combinations for Optimizing SCLC Treatment

Mounting evidence has demonstrated that targeting the key component of either the RSR pathway or DSB repair system as a monotherapy shows promising anti-tumor efficacy in SCLC. However, DDR-based targeted therapy in combination with other therapeutics may exhibit greater therapeutic effect in the treatment of SCLC.

### 7.1. Combination of Targeting RSR with Conventional Therapeutics

Given that long-term toxicity of chemotherapy to normal tissues, developing novel drug combinations will not only provide the most efficacious anti-tumor effects but also minimize drug toxicity in the process of cancer treatment. Recently, it was reported that CHK1 inhibition synergizes with cisplatin to induce substantial cell apoptosis in SCLC cells regardless of p53 status [95]. Meanwhile, preclinical studies indicated that PARP inhibitors potentiate the activity of chemotherapeutic agents [68]. In clinical trials, PARP inhibitor veliparib has been evaluated in combination with cisplatin or etoposide in extensive-stage SCLC, and this combination strategy has shown improved efficacy compared to the standard of care [23,24]. AZD1775 in combination with carboplatin are being investigated in clinic (NCT02937818). As radiotherapy is also the standard treatment option for SCLC patients, a recent study demonstrates PARP inhibition enhances the effects of radiotherapy in a panel of cell lines and patient-derived xenograft (PDX) models [96]. Moreover, combination of an ATR inhibitor with whole-brain radiation therapy in the treatment of SCLC is now under clinical investigation (NCT02589522).

### 7.2. Co-Targeting the RSR Pathway and the DSB Repair System

As SCLC may utilize multiple repair mechanisms to cope with replicative DNA damage in SCLC, targeting multiple components of a DNA damage response would hope to circumvent drug-resistance and eradicate this disease. A recent report demonstrates that WEE1 inhibition potentiates the activity of PARP inhibitor in a SCLC preclinical model [76]. A clinical trial of AZD1775 in combination with olaparib for relapsed SCLC is underway (NCT02511795). In addition to directly target these repair proteins, combining a PI3K pathway inhibitor and a PARP inhibitor has shown synergistic anti-tumor effects in SCLC and warrants further consideration for clinical trials [97]. Targeting epigenetic factors such as BET family members cause defects in DSB repair and might provide beneficial effects in combination with PARP inhibitors [98]. Preclinical studies using BET inhibitor and PARP inhibitor combination have shown the superior anti-tumor efficacy compared to either agent alone in various cancer types [99,100,101]. We speculate that BET inhibition might enhance the anti-tumor activity of PARP inhibitor in SCLC through interference with DNA damage response. 

### 7.3. Combination of Targeting DDR with Immunotherapy

Several preclinical studies have shown that the increase of replication stress triggers the activation of an innate immune response, providing a rationale for the combination of DDR-targeting agents and immunotherapy to treat SCLC [53,102,103]. Recently, the combination of chemotherapy with anti-PD-L1 demonstrated prolonged progression-free survival and was approved as a first-line treatment option for extensive-stage SCLC patients (NCT02763579) [104]. In another work, the authors elegantly showed that targeting PARP or CHK1 increases PD-L1 expression in SCLC cells and combined use of the PARP or CHK1 inhibitor with the PD-L1 blockade significantly augments anti-tumor efficacy compared to either monotherapy in SCLC [105].

## 8. Conclusions and Prospects

It has been increasingly apparent that exceptional high genome instability in SCLC is likely due to high levels of replication stress (Figure 1). SCLC cells count on an active RSR pathway and a constitutive DSB repair system to confront this genotoxicity for cellular survival. Exploiting the DNA damage response network provides an attractive avenue to treat SCLC. Besides the above targets, identification of novel targets that might interfere with either RSR or DSB repair pathways is worth further investigation. Targeted DDR therapy as a monotherapy or in combination with chemotherapy has shown promising anti-tumor effects in both preclinical and clinical studies. Nonetheless, the integration of targeted DDR therapy into other therapeutic strategies to achieve greater synergy effects and to overcome de novo resistance remains to be fully explored in SCLC.

Chromatin dynamics regulated by chromatin modifiers plays a crucial role in DNA replication and DNA repair. For instance, BET bromodomain family proteins, including BRD2, BRD3, BRD4, and BRDT, are well-recognized chromatin readers of histone acetylation marks. Several lines of evidence have indicated that BET family proteins are involved in DNA damage response. Inhibition of BRD4 attenuates a replication stress response and impairs HR repair, providing a rationale for combination of BET inhibitor with drugs targeting DDR for cancer treatment [106]. Many chromatin modifiers can be manipulated by small molecular inhibitors. Co-targeting epigenetic factors and DDR proteins is an area of interest in the improvement of the treatment efficacy for this recalcitrant disease. The links between metabolism and DNA damage response network has been well-documented in the past few years. Metabolic reprogramming is a common feature in SCLC [107]. Targets for regulating metabolism have the potential to leverage replication stress and DSB repair, thereby augmenting the therapeutic efficacy of DNA damage agents against SCLC. PARP inhibitors have been demonstrated to elicit an anti-tumor immune response, and PARP inhibition expands the efficacy of immune checkpoint blockade (ICB) in SCLC [105]. As PARP inhibition induces extensive replication stress, combination of other replication stress-inducing agents like the WEE1 inhibitor with ICB might also hold great promise to treat SCLC.

In summary, identifying novel targets and promising combinations between targeted DDR therapy and other therapeutics should pay more attention. Further investigations on the molecular mechanisms for the effects of single-agents and the synergy effects of combination treatments might improve the therapeutic efficacy for the treatment of SCLC.

## Figures and Tables

**Figure 1 cancers-11-01289-f001:**
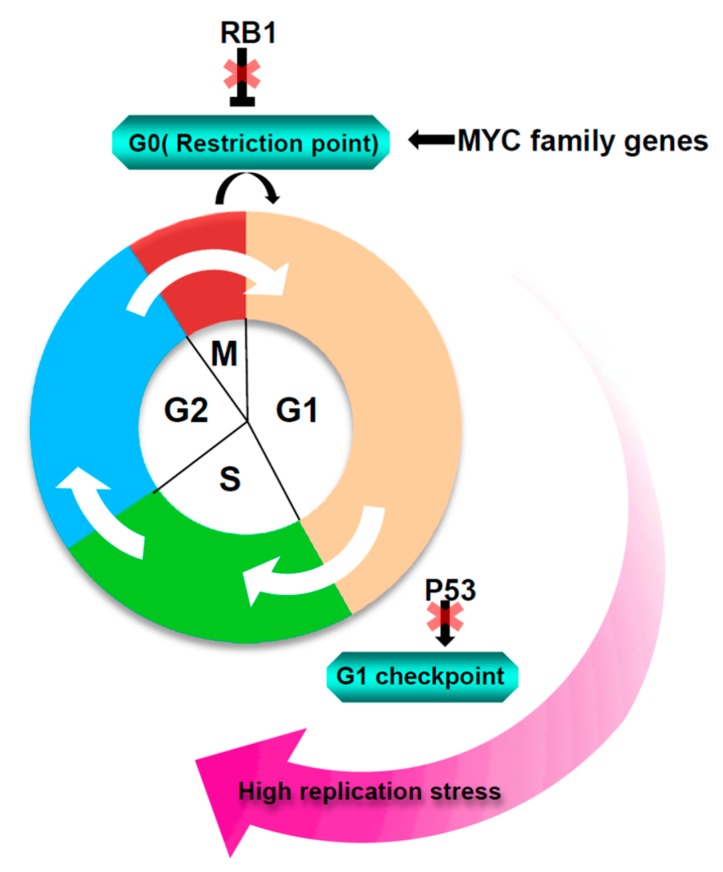
Causes of replication stress in small cell lung cancer (SCLC) cells. A number of endogenous obstacles that lead to replication stress in SCLC, including inactivation of tumor suppressors RB1 and p53, as well as amplification of *MYC* family genes. *RB1* loss or amplification of *MYC* family genes promotes G0-G1 cell cycle entry, *p53* mutation leads to deficient G1/S DNA damage checkpoint.

**Figure 2 cancers-11-01289-f002:**
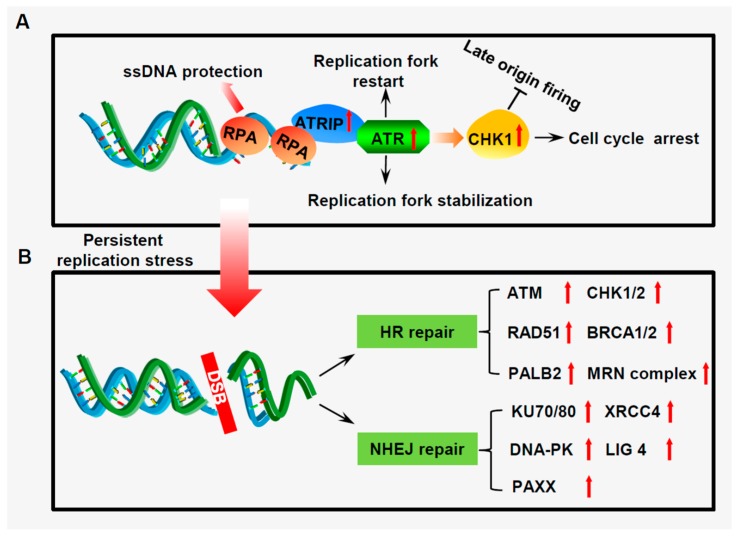
The DNA damage response network in SCLC cells. (**A**) The roles of the ATR-CHK1 signaling cascade in response to replication stress, replication protein A (RPA) is first loaded onto long stretches of single-strand DNA (ssDNA). The ATR was then recruited to the ssDNA sites through interaction with ATRIP. Once activated, ATR-mediated activation of CHK1 then transmits signals of replication stress through multiple mechanisms. (**B**) Once persistent replication stress cannot be fixed, replication forks then collapse into double-strand breaks (DSBs). In response to DSBs, SCLC cancer cells activate two major DSB repair pathways: Homologous recombination (HR) and non-homologous end joining (NHEJ). Crucial proteins involved in the HR repair pathway include BRCA1/2, CHK1/2, RAD51, ATM, etc.; the core components of the NHEJ repair pathway comprise KU70/80, XRCC4, DNA-PK, etc. Thus, constitutive activation of the replication stress response (RSR) pathway and DSB repair system is crucial for maintaining SCLC cell survival.

**Figure 3 cancers-11-01289-f003:**
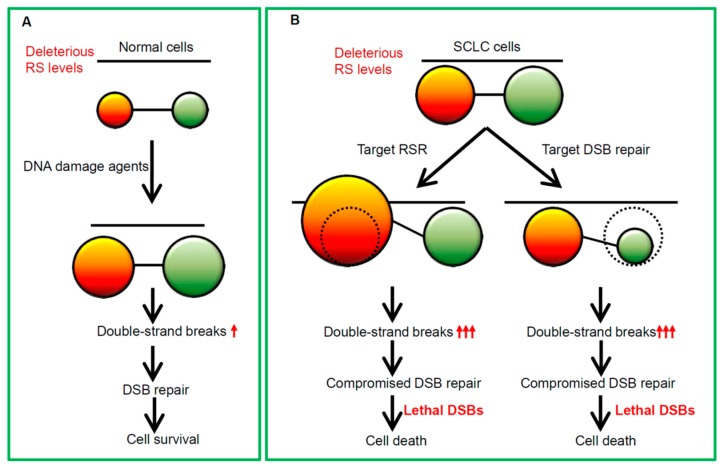
The rationales for targeting RSR and DSB repair in SCLC. (**A**) In normal cells, an elegant balance between the endogenous replication stress and competent DNA damage response pathway maintains genome integrity. (**B**) In SCLC cells, targeting RSR leads to intolerable replication stress (RS), thus leading lethal RS levels and DSBs that cannot be fixed by SCLC cells. Similarly, perturbation of DSB repair system results in compromised DNA damage repair and lethal DSBs that cannot be fixed by SCLC cells. Red arrow indicates an increase of DSBs.

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
