# Peer review of "Targeting DNA Replication Stress and DNA Double-Strand Break Repair for Optimizing SCLC Treatment"

_cancers, 2019, doi:10.3390/cancers11091289_

Round 1

Reviewer 1 Report

Comments to Authors.

The content of the review article is definitely one of the research hot areas.

Main comments:

These authors stated that: Here, we provide an overview of the major sources of RS in SCLC cells. Second, we outline the pharmacological approaches to exacerbate RS and to target DDR for SCLC TX … Lastly, we discuss and propose …

However, the three parts of the review article organization is not so clearly organized. Instead, they used the organization of: 1., 2., 3., 4., 5. (5.1., 5.2., 5.3.), 6. (6.1., …), 7. (7.1. ….), and 8. Conclusion and Prospects.

Many clinical studies are mentioned but the results are not presented to fill in readers’ curiosity and also for readers to evaluate relevant aspects they may have. For example, the Parp inhibitors’ case. Three trials mentioned but no further information of outcomes are provided. Although some may be due to not available but some mentioned good results came out but no results briefly presented to audience.

Of figure 3, it gives the expression of that when a cell has DSB, this cell will definitely only have a fate of death. But this may not true, unless there is no way to be repaired. However, DSB can be repaired as well in certain situations. So this figure needs to be revised to make it more complete to reflect the real situation. Normal cells never have DSB during DD agent treatment of cancer? If this is true, such DD agents should have no any side effects. I never hear such so good DD agents.

Minor Comments:

Need to carefully go through the manuscript to correct grammar, types, etc.. For example, in abstract, constitutively activation of …, P53 or p53? Use of P53 not p53 is unusual. I know we can use RB1 or Rb1.

Figure fonts should be larger when possible, especially Figures 2, 3.

A, B were missed in the Figure 3 image.

An abbreviation should always come out from the first time of the full name appearance. But this rule not followed at least in some cases. For example, Parp. Please check the entire article.

Author Response

Response to Reviewer 1 Comments:

Point 1: These authors stated that: Here, we provide an overview of the major sources of RS in SCLC cells. Second, we outline the pharmacological approaches to exacerbate RS and to target DDR for SCLC TX … Lastly, we discuss and propose …

However, the three parts of the review article organization is not so clearly organized. Instead, they used the organization of: 1., 2., 3., 4., 5. (5.1., 5.2., 5.3.), 6. (6.1., …), 7. (7.1. ….), and 8. Conclusion and Prospects.

Response 1: Thank you very much for the helpful comments. We re-wrote the paragraph to accurately describe how the article is organized as follow:

“Here, we provide an overview of the major sources of replication stress in SCLC cells and the signaling pathways responsible for DDR. We then outline the pharmacological approaches to exacerbate replication stress and to target DNA damage repair for SCLC treatment, with a specific focus on RSR and DSB repair. Lastly, we discuss the combination strategies for treatment and propose the potential strategies to further augment treatment efficacy for SCLC.”

Point 2: Many clinical studies are mentioned but the results are not presented to fill in readers’ curiosity and also for readers to evaluate relevant aspects they may have. For example, the Parp inhibitors’ case. Three trials mentioned but no further information of outcomes are provided. Although some may be due to not available but some mentioned good results came out but no results briefly presented to audience.

Response 2: Thank you for the constructive suggestion. We have added the results for several clinical studies that have been completed so far. As the reviewer realized, some clinical trials are underway and the clinical results have not been posted or published.

Point 3: Of figure 3, it gives the expression of that when a cell has DSB, this cell will definitely only have a fate of death. But this may not true, unless there is no way to be repaired. However, DSB can be repaired as well in certain situations. So this figure needs to be revised to make it more complete to reflect the real situation. Normal cells never have DSB during DD agent treatment of cancer? If this is true, such DD agents should have no any side effects. I never hear such so good DD agents.

Response 3: Thank you for the constructive suggestion. We have now modified the pictures in Figure 3 following the reviewer’s suggestion.

Point 4: Need to carefully go through the manuscript to correct grammar, types, etc.. For example, in abstract, constitutively activation of …, P53 or p53? Use of P53 not p53 is unusual. I know we can use RB1 or Rb1.

Response 4: We apologize for the misspelling for a certain gene and grammar errors. We have now corrected the writing errors in the revised manuscript.

Point 5: Figure fonts should be larger when possible, especially Figures 2, 3.

Response 5: Thank you for the helpful comments. We have resized the figure fonts as the reviewer suggested.

Point 6: A, B were missed in the Figure 3 image.

Response 6: Thank you for the helpful comments. We have now added missing information in Figure 3.

Point 7: An abbreviation should always come out from the first time of the full name appearance. But this rule not followed at least in some cases. For example, Parp. Please check the entire article.

Response 7: Thank you for the helpful comments. We have rechecked our manuscript to follow the common rule.

Reviewer 2 Report

Summary

The authors summarize components of the DNA replication stress response, DSB repair, and their role in SCLC. They also describe how these pathways are potential therapeutic targets for SCLC. This article is straightforward, well organized, and relatively easy to read. However I have two concerns with the manuscript in its current form. 1) There are many excellent reviews covering replication stress and DNA damage – this review does not offer much that is new. I highlight two points the authors could potentially expand on for a more exciting article. 2) There are a number of passages where the meaning of a sentence/passage is not clear. There are many grammatical errors (mostly agreement).

Main points

1) The authors mention their unpublished data in at least 2 instances where it would be interesting for them to expand on the implications by adding a specific example of the potential. The second is a new, interesting finding, and both are potential places to highlight their contributions (line 194, PARPi; line 329-330, BETi+PARPi method of action) to the well-known RSR/DSB repair story. The summary sentence in line 195 is generic, what would this mean for SCLC or what are the implications for new treatments?

*lines 332-339 - Conclusions: It would be interesting for the authors to discuss how epigenetic & metabolic pathways interact with the RSR and less on the well-documented RSR – DSB repair interaction. This will highlight their contributions and expand on an area warranting further study. Are these therapeutic strategies that are specific to SCLC? Could the authors’ SCLC studies apply more broadly to other cancer types? The metabolic pathways may be outside the scope of this review, but the BET inhibitor results feed directly into the epigenetic pathway interactions… It would be nice to read a paragraph on this.

2) There were multiple passages I found confusing. Below I highlighted some of the sections / sentences I found not clear. I did not include cases of grammar disagreement, etc as these were too frequent.

Detailed comments: Scientific or citation questions are marked with an *

- line 45-47 – I did not understand the sentence starting with “However there is still a fragile balance”

*line 193 – not sure what you mean by “augment” here… how does the PARPi help the RSR? does it increase or decrease the RSR?

- section 6: Grammar and agreement of sentences is way off. In particular, rewrite/simplify sentences #3 and #4 of this paragraph to clarify meaning (lines 247-251).

* Line 283 – amplification does not validate aurka as a treatment target – it is highlights it as a potential target.

* Line 285 – citation for loss of AURKA compromises resection?

Line 291 – more responsive than what? change: “significantly more responsive” à “responsive”

Line 298 – revise

Line 301-302 – Cut first sentence, redundant with previous sentence and the point is clear from the section title.

Line 309 – “some signal of efficacy”? what does this mean?

Section 7.2 – first two sentences say very similar things. Cut the redundancy. Use of “co-targeting” is not necessary, please use “targeting”.

*Line 344 – targeting the DDR is not unique – this is being done in many cancer types and SCLC is just another now. Also, existing chemotherapies already do this in a roundabout way by overwhelming the system. It’s a new target for SCLC?

Author Response

Response to Reviewer 2 Comments:

Main points

Point 1: The authors mention their unpublished data in at least 2 instances where it would be interesting for them to expand on the implications by adding a specific example of the potential. The second is a new, interesting finding, and both are potential places to highlight their contributions (line 194, PARPi; line 329-330, BETi+PARPi method of action) to the well-known RSR/DSB repair story.

Response 1: We thank the Reviewer for the thoughtful comments. We did discover the underlying mechanisms leading to the synergistic effect in SCLC cells treated by PARP inhibitor talazoparib in combination with BET inhibitor JQ1. We feel that it would be better not to discuss the results too much in this review article.

Point 2: The summary sentence in line 195 is generic, what would this mean for SCLC or what are the implications for new treatments?

Response 2: Thank you for the great suggestion. We have rewritten the sentence in line 195.

Point 3: *lines 332-339 - Conclusions: It would be interesting for the authors to discuss how epigenetic & metabolic pathways interact with the RSR and less on the well-documented RSR – DSB repair interaction. This will highlight their contributions and expand on an area warranting further study. Are these therapeutic strategies that are specific to SCLC? Could the authors’ SCLC studies apply more broadly to other cancer types? The metabolic pathways may be outside the scope of this review, but the BET inhibitor results feed directly into the epigenetic pathway interactions… It would be nice to read a paragraph on this.

Response 3: Thank you for the helpful comment. We have expanded several sentences to discuss how epigenetic modulators interact with RSR in section 8. We used BET family proteins as examples to show how chromatin modifiers are involved in RSR and DSB repair, which form the basis to co-targeting chromatin modifiers and either RSR or HR pathway for cancer treatment. We should mention that future studies are needed to understand the underlying mechanisms that how BET proteins and other chromatin modifiers interfere with RSR pathway.

Of note, we think that these therapeutic strategies are not limited to SCLC and tumors that with high endogenous replication stress may be highly responsive to these therapeutic strategies, too.

Point 4: There were multiple passages I found confusing. Below I highlighted some of the sections / sentences I found not clear. I did not include cases of grammar disagreement, etc as these were too frequent.

Response 4: We have rechecked the manuscript and corrected the grammar errors as much as we can. We believe that the revised manuscript is much improved thanks to your helpful comments.

Detailed comments: Scientific or citation questions are marked with an *

Point 5: line 45-47 – I did not understand the sentence starting with “However there is still a fragile balance”

Response 5: Thank you for the helpful comment. We have now rewritten the sentence to make it easily readable in the revised manuscript.

Point 6: *line 193 – not sure what you mean by “augment” here… how does the PARPi help the RSR? does it increase or decrease the RSR?

Response 6: Thank you for the helpful comment. We have now corrected it into “PARP inhibitor talazoparib remarkably increases replication stress response in SCLC cells” in the revised manuscript.

Point 7: - section 6: Grammar and agreement of sentences is way off. In particular, rewrite/simplify sentences #3 and #4 of this paragraph to clarify meaning (lines 247-251).

Response 7: Thank you for the helpful comments. We have reorganized the sentences #3 and #4 to make it more readable.

Point 8: * Line 283 – amplification does not validate aurka as a treatment target – it is highlights it as a potential target. Aurora kinase a – normally referred to as aurora A? I don’t think so….

Response 8: Thank you for the helpful comments. We have rewritten the sentence and corrected the gene name and protein description for Aurora kinase a in section 6.

Point 9: * Line 285 – citation for loss of AURKA compromises resection?

Response 9: Thank you for the helpful comments. We have added the missing reference in the revised manuscript.

Point 10: Line 291 – more responsive than what? change: “significantly more responsive” à “responsive”

Response 10: Thank you for the helpful comments. We have changed the sentence to “Further investigation indicates that SCLC cells with MYC amplification or MYC overexpression are significantly more responsive to Aurora kinase inhibitors such as alisertib and barasetib than MYC family gene-nonamplified SCLC”.

Point 11: Line 298 – revise

Response 11: Thank you for the helpful comments. We have rewritten this sentence in the revised manuscript.

Point 12: Line 301-302 – Cut first sentence, redundant with previous sentence and the point is clear from the section title.

Response 12: We apologize for the redundancy in writing and have now cut the first sentence in the revised manuscript.

Point 13: Line 309 – “some signal of efficacy”? what does this mean?

Response 13: We apologize for the misleading writing and have now corrected the sentence in the revised manuscript.

Point 14: Section 7.2 – first two sentences say very similar things. Cut the redundancy. Use of “co-targeting” is not necessary, please use “targeting”.

Response 14: Thank you for the helpful comments. We have cut the redundant sentence and have changed the “co-targeting” into “targeting” in the revised manuscript.

Point 15: *Line 344 – targeting the DDR is not unique – this is being done in many cancer types and SCLC is just another now. Also, existing chemotherapies already do this in a roundabout way by overwhelming the system. It’s a new target for SCLC?

Response 15: Thank you for the helpful comments. We have corrected the writing in the manuscript.